# Aggressiveness in Italian Children with ADHD: MAOA Gene Polymorphism Involvement

**DOI:** 10.3390/diseases12040070

**Published:** 2024-03-31

**Authors:** Ludovico Neri, Beatrice Marziani, Pierluigi Sebastiani, Tiziana Del Beato, Alessia Colanardi, Maria Pia Legge, Anna Aureli

**Affiliations:** 1Neurology and Psychiatry Unit for Children and Adolescents, San Salvatore Hospital, via L. Natali, 1, Coppito, 67100 L’Aquila, Italy; ludovico.neri@graduate.univaq.it (L.N.); mlegge@asl1abruzzo.it (M.P.L.); 2Emergency Medicine Department, Sant’Anna University Hospital, Via A. Moro, 8, Cona, 44124 Ferrara, Italy; mrzbrc@unife.it; 3CNR Institute of Translational Pharmacology, Via Carducci 32, 67100 L’Aquila, Italy; pierluigi.sebastiani@cnr.it (P.S.); tiziana.delbeato@cnr.it (T.D.B.); alessia.colanardi@cnr.it (A.C.)

**Keywords:** ADHD, aggressive behavior, MAOA, 5-HTT

## Abstract

ADHD is a neurodevelopmental disorder that children and adults can develop. A complex interplay of genetic and environmental factors may underlie interindividual variability in ADHD and potentially related aggressive behavior. Using high-resolution molecular biology techniques, we investigated the impact of some MAOA and SLC6A4 variations on ADHD and aggressive behavior in a group of 80 Italian children with ADHD and in 80 healthy controls. We found that homozygous genotypes of MAOA rs6323 and rs1137070 were associated with an increased risk of ADHD (*p* = 0.02 and *p* = 0.03, respectively), whereas the heterozygous genotypes (GT of rs6323 and CT of rs1137030) (*p* = 0.0002 and *p* = 0.0006) were strongly linked to a lower risk of developing this disorder. In patients with aggressive behavior, we highlighted only a weak negative association of both MAOA polymorphisms (heterozygous genotypes) with aggressiveness, suggesting that these genotypes may be protective towards specific changes in behavior (*p* = 0.05). Interestingly, an increase in the GG genotype of rs6323 (*p* = 0.01) and a decrease in GT genotype (*p* = 0.0005) was also found in patients without aggressive behavior compared to controls. Regarding 5HTT gene genotyping, no allele and genotype differences have been detected among patients and controls. Our work shows that defining a genetic profile of ADHD may help in the early detection of patients who are more vulnerable to ADHD and/or antisocial and aggressive behavior and to design precision-targeted therapies.

## 1. Introduction

### 1.1. Characteristics and Prevalence of ADHD

Attention-deficit/hyperactivity disorder (ADHD) is a clinically heterogeneous condition due to both genetic and environmental factors and is highly hereditary and associated with different brain abnormalities [1,2]. The global prevalence of ADHD in children and adolescents is 7.6% in children aged 3 to 12 years and 5.6% in teenagers aged 12 to 18 [3], while, in Italy, it is estimated to be between 1.1% and 3.1% in individuals aged 5–17 years [4].

Main ADHD symptoms involve inattention, hyperactivity, and/or impulsivity and are more frequently observed in males than in females (gender ratios from 3:1 to 9:1) [5,6]. However, it should be noted that ADHD is not a predominantly male disorder and that, at least in part, the gender difference described may depend on the fact that the majority of children referred to the clinic are males [7].

ADHD is typically complicated by extensive comorbid conditions, and, often, it co-occurs with autism spectrum disorder (ASD) despite the differences in their core symptoms and diagnostic criteria [8]. It seems that the ADHD–ASD phenotypic overlap may be partially explained by shared genetic influences [9]. In support of this, Gidziela et al. reported that most studies included in their metanalysis examined and confirmed the genetic correlations between ASD and ADHD. They also highlighted that other neurodevelopmental disorders beginning in childhood, such as impulse control and conduct disorders, show similar features to that of impulsive behavior, aggressiveness, and pathological rule breaking [10]. Nevertheless, studies on the association between these disorders are still few and heterogeneous; therefore, more developmental genetics research are needed to provide new insights to guide potential clinical and educational diagnostic procedures and practice.

### 1.2. Factors Associated with ADHD

As with most complex diseases, ADHD onset is due to the contribution of several kinds of factors.

On the one hand, environmental factors play a crucial role on the presentation and the characteristics of the disease. For example, it is interesting to note that during the COVID-19 pandemic, a significant increase in ADHD cases was recorded. It is possible that COVID-19-related stressors triggered or aggravated cognitive, emotional, and physiological reactions that lead to impulsive–aggressive behavior [11].

On the other hand, a genetic component has been amply documented in the etiology of ADHD. Genes encoding the dopamine receptors (e.g., DRD2 and DRD4), dopamine transporter (DAT1), and monoamine oxidase (MAOA) enzymes, such as catechol-O-methyltransferase (COMT) and dopamine beta-hydroxylase (DBH), are the most studied [12,13,14]. However, the influence of genetics on disease risk needs to be better understood [15]. Children with ADHD can also exhibit hostility or anger and aggressive behavior. Indeed, impulsive aggression is often described among comorbidities in individuals with ADHD [16,17]. There are several biologically interesting markers that play a key role in the neurophysiological response to stressful life events [18]. Among these, MAOA, which encodes the key enzyme for the degradation of serotonin and catecholamines, is the best-documented gene. Human genetic studies associated a higher risk for antisocial and violent behavior both with congenital MAOA deficiency and low-activity MAOA variants [19]. Moreover, a sexual dimorphic effect of MAOA in behavioral traits beyond an increase in its activity levels with age has been detected [20]. Studies indicated that various MAOA polymorphisms are involved in ADHD and/or traits of aggressiveness. Most prominently, rs6323 and rs1137070 polymorphisms, respectively located in exon 8 and exon 14 of the MAOA gene seem to be associated with ADHD and/or personality traits of aggressive behavior, impulsivity, and anti-social behavior [21,22,23,24,25]. Also, the serotonin transporter gene (SLC6A4*)* encoding for the serotonin transporter protein (5-HTT) has been associated with impulsive–aggressive behavior, although with discordant results that reflect the importance of the environmental influence. Within the promoter region (5-HTTLPR) of the 5-HTT gene, there is a 44-bp insertion/deletion variable-number tandem repeat polymorphism that generates two allelic forms: the long (l) and the short (s) variants; the first one has a more transcriptional activity and higher serotonin uptake than the second one [26]. Some studies highlighted a positive association between the l-allele and violence or aggression [27,28], while others found an excess of the s-allele among violent males [29,30]. Given the multiple factors that contribute to aggressive behavior development, understanding every gene contribution to aggressiveness is crucial.

### 1.3. Aim and Scope

After the COVID-19 pandemic, in our geographical area, we have documented an increase in cases of neurodevelopmental disorders such as ADHD, which also happened after the violent earthquake of 2009 [31]. Children diagnosed with ADHD admitted to our hospital often showed difficulties in socialization, low self-esteem, and poor interpersonal skills. Sometimes, they developed an aggressive behavior. Therefore, to deepen our understanding on the genetics of aggressiveness, we wanted to investigate the influence of MAOA and SLC6A4 gene polymorphisms on aggressive behavior in a group of Italian children with ADHD. Specifically, we provide a novel insight on how to define the risk level of impulsive/aggressive personality traits through typing of a group of genetic polymorphisms. Moreover, our results might provide a useful tool for the right therapeutic treatment.

## 2. Materials and Methods

### 2.1. Population Sample and Diagnostic Assessment

This case-control study was conducted on a total of 80 subjects with ADHD, aged between 6 and 17 years, who were selected among the individuals visited at the Unit of Child Neuropsychiatry at San Salvatore Hospital in L’Aquila, Italy. Demographic characteristics are shown in Table 1. The diagnosis of ADHD (inattentive, impulsive-hyperactive, and combined) was determined. First, a complete medical anamnesis was collected and then, the presence of symptoms of ADHD in different life settings like home, school and social ones, according to DSM-V, was assessed and the symptoms were quantified. Successively, general objective and neurological objective exams were performed, and all these data were registered. For suspected cases of ADHD, the confirmation of the diagnosis and its characterization were obtained with the scores calculated from the analysis of the answers to the Conners’ Rating Scales—III Edition questionnaire completed by caregivers. This clinical scale was also useful to confirm the level of impairment consequent to the presence of the symptoms of the disorder. Aggressive behavior was first investigated in the anamnestic collection with dedicated questions and then evaluated using the subscale Defiance/Aggression contained in the Conners’ Rating Scales—III Edition. This is a score used specifically to identify aggressive behavior and to characterize its relevance. We considered a subject as aggressive when the score in this subscale was over 2 standard deviations with respect to the average of the population and when the presence of such behavioral traits was responsible for a function impairment. All the considered scores were evaluated with regard to the interview validity indexes according to rules contained in the manual for the correct interpretation of the results. The mean values of the Conner’s Rating Scales in the total sample group, including boys and girls, are shown in Appendix A. The exclusion criteria were autism spectrum disorder, intellectual disability, or previously diagnosed genetic disorders. Eighty unrelated individuals without ADHD were used for comparison. This study was performed in accordance with the standards of the Ethics Committee (Code 0102550), and informed consent was obtained from all participants in this study or from their legal guardians.

### 2.2. Genotyping

Genomic DNA was obtained from peripheral blood (PB) cells using the QIAamp DNA Blood Mini Kit (Qiagen, Hilden, Germany). This procedure permits a fast and efficient genomic DNA purification from medium volumes of human sample blood. A brief description of the method is described here. After a first phase of lysis, the lysate is loaded onto the spin column, and while the DNA binds to the QIAamp membrane, impurities are washed away in two centrifugation phases. Finally, the DNA is eluted and stored at −20 °C until use. We determined the DNA purity and concentration by a spectrophotometer (Beckman Instruments, Fullerton, CA, USA). The genotypes of the two selected exonic SNPs of the MAOA gene (rs6323, Arg297Arg and rs1137070, Asp470Asp) were analyzed by direct sequencing using primers shown in Table 2.

PCR amplifications were carried out in a total amount of 50 μL (reaction mixture), including 50 ng of genomic DNA, 1× PCR buffer with MgCl_2_, 0.2 mM each of deoxynucleotides, 0.5 U of Taq polymerase, and a 3.2 pmol/μL concentration of each primer. The PCR process consisted of three main steps: an initial denaturation at 95 °C for 5′, followed by 35 cycles of amplification at 94 °C for 30 s, 57 °C for 45 s, and 72 °C for 30 s and finally, an elongation at 72 °C for 7 min. PCR products were electrophoresed on a ethidium bromide-stained 1.5% agarose gel and subsequently purified by a PCR clean-up reagent (EXOSAP, Wiesbaden, Germany). Sequence reactions were performed using the Big Dye Terminator Chemistry v 1.1 (Applied Biosystems, Foster City, CA, USA), then processed on a 16-capillary sequencing instrument (ABI Prism 3130 Genetic Analyzer, Applied Biosystems), and finally, purified.

All data were collected, and typing was acquired by an alignment of the processed sequences with those of exon 8 and exon 14 of the human MAOA gene retrieved from the GenBank. The 5-HTTLPR gene polymorphism study was performed with PCR using the following primers: forward 5′-GGCGTTGCCGCTCTGAATGC-3′ and reverse 5′-GAGGGACTGAGCTGGACAACCAC-3′. PCR cycling parameters included an initial denaturation phase at 94 °C for 5 min, followed by 35 cycles at 94 °C for 30 s (template denaturation), 64 °C for 30 s (primer annealing), and 72 °C for 60 s (primer extension), with a final extension at 72 °C for 7 min and a holding step with a temperature of 4 °C. The PCR products were identified on 2% agarose gel, stained with ethidium bromide. Illumination with ultraviolet light allowed to visualize bands and differentiate the 5HTT long allele (l) consisting of 528 base pair from the short one (s, 484 bp).

### 2.3. Statistical Analysis

Allele and genotype frequencies were obtained by direct counting. Pearson’s chi-square test or Fisher’s exact test were used to identify allele and genotype frequency differences between ADHD individuals and controls. Bonferroni’s correction (p_c_) was applied as appropriate. The Hardy–Weinberg equilibrium (HWE) of alleles was applied to assess population stratification and verify the hypothesis of non-random mating. The odds ratio (OR) and 95% confidence interval (CI) were estimated to calculate the risk of disease with respect to a particular genotype. A *p*-value of 0.05 or lower was considered to be significant. The specific sample size calculation was carried out a priori; 138 observations were required to obtain a power of 95% or higher in the chi-square test between patients and controls. Our study was conducted on 180 individuals and thus, the association of considered SNPs and ADHD can be sufficiently accounted for the results in this research. In accordance with the purpose of this study, regression analyses were conducted to evaluate the effects of rs6323 and rs1137070 of the MAOA gene and rs4795541 of the SLC6A4 gene on ADHD and/or aggressive behavior. SPSS statistics software (version 21) was used for data and statistical analysis.

## 3. Results

The study group was composed of 80 patients with ADHD (74 males and 6 females, with a mean age of 10.8 years), of which 3 were predominantly hyperactive-impulsive, 9 were predominantly inattentive, and 68 were both inattentive and hyperactive–impulsive (combined, according to the DSM-V categorization). The 80 cases included in this study were further stratified into two subtypes based on the presence (n = 32, 40%) or absence (n = 48, 60%) of aggressive behavior. Eighty sex- and age-matched healthy individuals were evaluated as controls. The genotype/allele frequencies of the two *SNPs* (*rs6323* and *rs1137070*) of MAOA in cases and controls are summarized in Table 3. We found that the GG genotype frequency of rs6323 was significantly higher in the ADHD group than in the control group (*p* = 0.02, OR = 3.00, 95% CI: 1.23–7.30). However, the GT genotype was associated with significantly lower risk of ADHD (*p* = 0.0002, OR = 0.15, 95% CI: 0.05–0.45). This result has been validated by Bonferroni’s correction for multiple comparison and the *p*-value remained significant (p_c_ = 0.0006), suggesting that the GT genotype would be a biomarker of resistance to ADHD. The analysis of the rs1137070 polymorphism revealed that the TT genotype was associated with significantly higher risk of ADHD (*p*= 0.0344, OR = 2.49, 95% CI: 1.05–5.91), while the presence of the CT genotype was negatively associated with the development of this disorder (*p* = 0.0006, OR = 0.19, 95% CI: 0.07–0.53). OR indicates that the presence of this genotype drops the risk for ADHD, and the *p*-value remained significant (p_c_ = 0.002) after Bonferroni’s correction.

Interestingly, the positive association with ADHD was also confirmed when the predisposing genotypes of the two MAOA gene polymorphisms (GG of rs6323 and TT of rs1137030) were considered together (23.7% (19/80) vs. 10% (8/80) *p* = 0.0238, OR = 2.80, CI: 1.15 to 6.85). In addition, the presence of both genotypes negatively associated with ADHD (GT + CT) underscored the reduced risk to develop ADHD (5% (4/80) vs. 25% (20/80) *p* = 0.0004, p_c_ = 0.0012, OR: 0.16, CI = 0.05 to 0.49). Therefore, we also investigated if MAOA was associated with aggressive behavior. Considering the subtype of patients with aggressive behavior, our findings highlighted the same trend observed in the total group of patients with a prevalence of the TT genotype of rs6323 and CC genotype of rs1137070 (Table 4). However, only a weak negative association was found between both MAOA polymorphisms and aggressiveness; the GT genotype of rs6323 and CT genotype of rs1137070 were less frequent in patients with aggressiveness than in controls, suggesting that these heterozygous genotypes may be protective towards specific changes in behavior, namely aggression (9% vs. 26%, *p* = 0.05, OR = 0.30, CI: 0.08 to 1.07). Furthermore, we found that the GG genotype of rs6323 was significantly increased among patients without the aggressive component compared to controls (27% vs. 10%, *p* = 0.01, p_c_ = 0.03, OR = 3.34, CI: 1.27 to 8.81) and the GT genotype was instead less frequent (2% vs. 26%, *p* = 0.0005, p_c_ = 0.0015, OR = 0.06, CI: 0.01 to 0.46). Additionally, we found that the CT genotype of rs1137070 was less frequent in this group of patients than in controls (6.2% vs. 26%, *p* = 0.0055, OR = 0.19, CI: 0.05 to 0.68).

Also, a significantly greater combination of rs6323 GG and rs1137070 TT genotypes was observed in the subtype of patients without aggressive behavior compared to controls (25% (7/32) vs. 10% (8/80) *p* = 0.02, OR = 3.00, CI: 1.13 to 7.99). Moreover, the 2-marker haplotype analysis indicate a protection profile to ADHD, in which the rs6323 GT and rs1137070 CT genotypes carriers (2% (1/48) vs. 25% (20/80) *p* = 0.0007, OR = 0.06, CI: 0.01 to 0.49) have less chance to display an aggressive behavior. No statistically significant differences in MAOA allele frequencies were detected between patients and healthy individuals. All study participants were also genotyped based on the long and short alleles of the 5HTT gene, but no allele and genotype differences were detected among patients and controls. The distribution of genotypes in the 5-HTT was as follows: 14 patients (17.5%) and 16 controls (20%) had two short alleles (ss), 22 patients (27.5%) and 16 controls (20%) had two long alleles (ll), and 44 patients (55%) and 48 controls (60%) had one long allele and one short allele (ls). Lastly, no associations between aggressive behavior and the 5-HTTLPR genotypes were found, as shown in Table 5.

Then, regression analyses with the main effects of the genotypes were conducted; we observed a main effect of the GT genotype of rs6323 of the MAOA gene, which negatively predicted ADHD (*p* = 0.04).

Both controls and patients were not in Hardy–Weinberg equilibrium at each locus investigated, thus indicating that the population experienced evolutionary changes, probably due to factors such as genetic drift, gene flow, natural selection, mutation, and non-random mating.

## 4. Discussion

In the last few years, there has been a substantial increase in social anxiety, paranoia, poor interpersonal relationships, and aggressive behavior among young patients admitted to our local unit of Child Neuropsychiatry, which led us to conduct this study. The starting point was the analysis of the genetic contribution of the MAOA gene, also known as the “killer gene”, to ADHD onset and especially, aggressive behavior development in our group of patients. Our results demonstrated the existence of an association between the SNPs rs6323 and rs1137070 of MAOA and this disorder. In addition, we were interested in verifying if the 5-HTT gene polymorphism also had a role in the complex etiology of ADHD, but, in this case, we failed to find significant results. We are aware that the understanding of ADHD requires a multidisciplinary approach and that imbalances in biological, psychological, and environmental factors are at the basis of aggressive behavior onset. Nonetheless, scientific research is making great progress on the comprehension of neural mechanisms involved in impulsive aggression [32].

Mutations of several genes, certain mental health conditions, and/or stress, fear, or a sense of losing control may also generate aggression (Figure 1) [33].

In children and adolescents, it is also more complex to understand the mechanisms underpinning aggression, given their difficulty in expressing emotions in words to communicate restlessness, anxiety, or frustration. Neurodevelopmental disorders, such as ADHD, can also play a part in aggressive behavior [16]. In most cases, this behavior is triggered by a particular event, such as a pandemic [34]. Aggressiveness can happen as a natural response to pandemic-related stressors. During the past devastating COVID-19 outbreak, the priority was to develop treatment options to control or prevent the spread of the virus. However, afterward, in that scenario, clinicians realized the increasing need to investigate and manage the wide range of COVID-19 complications, including mental health issues [35]. Italy was one of the most affected countries, and the government soon decided to apply lockdown measures to limit new epidemic waves [36]. Therefore, the limitations due to this global health crisis also had a strong negative impact on the mental health of children and adolescents and caused an increase in mood disorders, eating disorders, sleep disorders, and loneliness or social withdrawal, as well as the appearance of neurodevelopmental disorders such as ADHD. It is interesting to note that the perception of a greater health risk, the fear of an unexpected situation, together with social isolation have often triggered reactions of stress and discomfort not only in affected people but even in the general population [37].

With regard to the genetic component, scientists have found a complex genetic architecture of aggressive behavior in the context of ADHD, and several potential risk genes have been identified [38]. Among these, dopamine and serotonin, key neuromodulators involved in aggressive behavior, have received substantial attention, even if the results are unsuccessful [39,40]. Nonetheless, a clear relationship between the MAOA and SLC6A4 genes and aggressiveness has been shown [41]. Indeed, it is known that MAOs have a key role in the metabolism of neurotransmitters (e.g., serotonin) involved in aggressive behavior [42,43]. Monoamines are widely expressed in the CNS. According to the current scientific literature, we focused on the serotonergic system that develops through the mesencephalic raphe nuclei to the frontal cortex and reaches different areas, especially those belonging to the limbic system, such as the cingulate cortex, amygdala (crucial area for emotional regulation), and hippocampus [44], which are the basis for many processes like motivation, attention, anxiety, irritability, and emotional and behavioral regulation [45]. In fact, the limbic system represents the physiological basis for the regulation of emotions and behavior as it is the physiological and psychological system that elaborates the emotions and the related vegetative responses. It is recognized that the reduction of serotonergic transmission may increase impulsive–aggressive behavior in different situations, such as those in which genetic variability and environmental factors intervene. For example, it has been reported that being exposed to repeated violent acts (environmental factor) may induce plastic modifications in the raphe serotonergic cells, resulting in altered serotonergic transmission and, consequently, an altered functioning of the prefrontal cortex [46,47] and an impaired capacity to regulate the mood, aggressiveness, impulsivity, and emotions. Interestingly, the MAOA enzyme is the first one that is responsible for the degradation of serotonin, and another key enzyme for the regulation of synaptic serotonin levels is SERT, codified by the SLC6A4 gene. Many studies investigated the relationship between the aggressive phenotype and the MAO system and have found that specific allelic variations in the genes encoding MAO are associated with aggressive traits [48]. In 1993, aggressive phenotypes of the Brunner syndrome were found to be related to mutations in the MAOA gene, which led to a complete or partial loss of the MAOA enzyme activity [49]. More recently, it has been understood that these mutations increase the activity of dopaminergic neurons through the upregulation of the N-methyl-D-aspartate receptor (NMDAR) function, and this might explain impulsivity and the maladaptive behavior associated with the Brunner syndrome [49].

The rs6323 variant of MAOA, located in the exonic region, is associated with altered enzyme activity; the G allele is linked to higher levels of the enzyme and the T allele is linked to lower levels [50]. Some research groups have reported that the low-activity MAOA genotype (TT) is associated with more aggressive behavior [51], while others have reached the opposite conclusion, identifying the high-activity MAOA genotype (GG) as the one associated with more aggressive behavior [52].

Regarding rs1137070, it has been shown that the T variant is associated with higher activity than that of the C allele [50]. Moreover, it is considered a useful biomarker of attentional problems; in this regard, the study by Lundwall et al. showed that this SNP predicted poorer developmental course in reaction time (RT) and that in girls, the carrier of CC genotype led to an increase in RT on a reflexive attention task between infancy and childhood [53].

Based on these observations, we investigated genetic variations in MAOA that underlie aggressive phenotypes in the ADHD context. Our study highlighted a different distribution of both MAOA rs6323 and rs1137070 genotypes between ADHD patients and healthy controls. The homozygous genotypes (GG and TT, respectively) were significantly associated with ADHD risk. Conversely, heterozygous genotypes (GT and CT) were associated with a lower risk of ADHD development. Furthermore, a detailed study of rs6323 and rs1137070 genotype combinations showed that the combination of rs6323 GG and rs1137070 TT genotypes has an additional predisposing effect on the development of ADHD, while that of heterozygous (GT and CT) genotypes shows a protective association toward the disorder. When we performed subtype analysis (aggressive behavior/non-aggressive behavior) of ADHD, we found a protective association between heterozygous genotypes of both SNPs of the MAOA gene and aggressive subtype. Moreover, the GG genotype of rs6323 was a risk factor for the development of aggressive behavior in non-aggressive patients. Instead, the GT genotype of rs6323 and CT genotype of rs1137070 were protective toward aggressive symptoms of ADHD. This finding could appear opposite to the expected one; however, in support of this, Zhao et al. demonstrated that individuals carrying the G allele seem to be more easily affected by the environment than those carrying the T allele. They also reported that adolescents with the G genotype may be more sensitive to certain environmental situations because they recall emotion-related brain regions and neurophysiological responses over activation [54]. Furthermore, Johnson et al. suggested that exposure to different environments alters the genetic vulnerability of adolescents by influencing the expression of genetic factors [55]. In this regard, it is of interest that a significantly positive relationship between exposure to COVID-19 and an increased risk of stress has been detected and that this can lead to emotional/behavioral outcomes, such as anxiety, which might represent a mediating factor for aggressive behavior development [11].

In addition, since an interactive effect of MAOA and 5-HTT gene polymorphisms on brain activity has been previously reported [56] and given the strong allele–allele interaction in the anterior cingulate cortex, we also wanted to investigate if the polymorphism in the promoter region of the serotonin transporter (5HTT) was implicated in impulsive/aggressive behavior in our study group.

Indeed, as already reported by Cunha-Bang and Knudsen, in vivo neuroimaging studies have shown alterations in serotonin receptors, indicating that low levels of synaptic serotonin are associated with high levels of impulsive aggression, and imaging genetics demonstrated that serotonergic genetic polymorphisms are associated with antisocial behavior [32].

However, no differences in 5-HTT genotype/allele frequencies among patients and controls have been found in our study, therefore, at the moment, we cannot confirm this hypothesis.

It should be kept in mind that there are differences in findings, which may be explained by gene–environment interactions, and further investigations are needed to clarify the role of the serotonin transporter in the etiopathogenesis of ADHD and/or in the severity of hyperactivity/impulsivity.

## 5. Conclusions

The complex etiology of ADHD is still not fully understood. Sometimes, ADHD symptoms are linked with physical comorbidities, such as asthma in early childhood, injuries, sleep disturbances, epilepsy, and excess weight, which can mask early diagnosis. It follows that a late diagnosis of cognitive and functional defects associated with ADHD can lead to adults with social and interpersonal difficulties. Understanding how patients with ADHD respond to specific environmental changes and identifying genetic markers associated with phenotypic plasticity might help to create and validate new tools, such as algorithms [57] that are able to identify ADHD in children, classify them, and distinguish those who are at highest risk for aggressive behavior. Although, as listed above, environmental conditions are of considerable importance in the development of this mental disease, the study of the genetic component involved is becoming increasingly important. In fact, the ability of genetics to predict the risk of developing a particular disease has now been demonstrated. Therefore, based on this assumption, we evaluated the importance of specific polymorphisms of the MAOA gene as susceptibility biomarkers of ADHD and/or aggressive traits. These are able to modify the functioning of the monoaminergic system and are, consequently, responsible for an altered monoaminergic transmission in different brain areas, such as those belonging to the limbic system. It is reasonable that these alterations have repercussions on brain activity, which can be registered via electroencephalography in areas like the frontal lobe. Different studies have already demonstrated deficits in neuropsychological mechanisms like spatial organization and behavioral modulation in subjects with ADHD [58]. So, it would be interesting to compare genetic findings to neurophysiological data to obtain a progressively more complete profile of ADHD necessary for a tailored therapy according to the rules of precision medicine. Such an approach could help to define a risk profile for ADHD based on both genetic and neurophysiological markers crucial to early intervention.

However, considering that it is not always easy to use genetic information to make clinical decisions, especially in diseases like ADHD, which are characterized by an extremely wide genetic component, more studies are needed in order to use this information in everyday clinical practice.

## Figures and Tables

**Figure 1 diseases-12-00070-f001:**
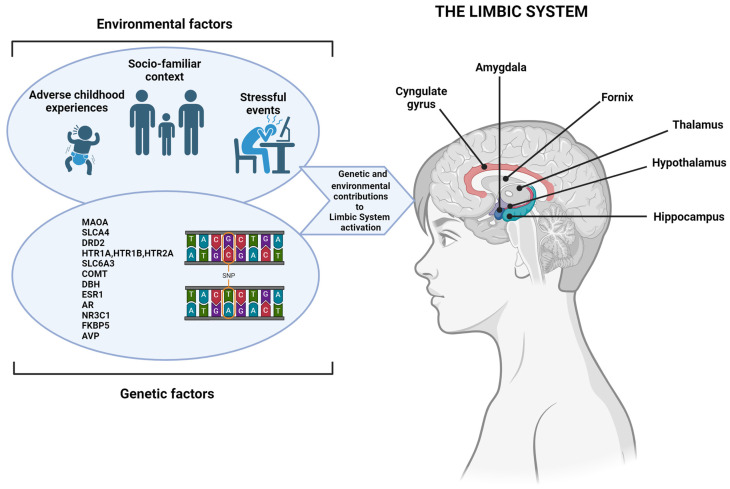
Multiple genetic variants are involved in ADHD development. As in most psychiatric diseases, genetic and environmental factors precociously affect the development of the central nervous system. The interaction between genetic predisposition and the environment, including adverse events that occur in the individual’s life (so-called stressors), may induce certain behaviors or psychological disorders. The relationship between genetic and environmental risk factors that may underlie interindividual variability and susceptibility to ADHD and/or impulsive aggressive behavior are shown here. The Figure was created with Biorender.

**Table 1 diseases-12-00070-t001:** Demographic characteristics of ADHD group (n = 80).

**Age** (Mean ± SD)	10.8 ± 0.84
**Sex** (n, %)	
Female	6 (7.5%)
Male	74 (92.5%)

**Table 2 diseases-12-00070-t002:** PCR and Sequencing primers for MAOA gene.

**MAOA rs6323** **F**	5′-TAATTAATGCGATCCCTCCG-3′
**MAOA** **rs6323 R**	5′-TGAGGAAATTGACAGACCAAGA-3′
**MAOA rs1137070** **F**	5′-GGCAACGTTTTTGGCATCTGGTC-3′
**MAOA** **rs1137070 R**	5′-ACTCATGCTGACAAGGAGGAACA-3′

**Table 3 diseases-12-00070-t003:** Allele and Genotype Frequencies of rs6323 andrs1137070 Polymorphisms of MAOA Gene in ADHD Patients and Controls.

	ADHD		Controls		*p*	OR
**MAOA SNP rs6323 (G891T)**						
**Genotypes**	n = 80	%	n = 80	%		
TT	56	0.70	51	0.64	ns ^§^	-
GT	4	0.05	21	0.26	0.0008	0.1479
GG	20	0.25	8	0.10	0.02	3.0
**Alleles**	2n = 160	%	2n = 160	%		
T	116	0.73	123	0.77	ns ^§^	-
G	44	0.27	37	0.23	ns ^§^	-
**MAOA SNP rs1137070 (T1410C)**						
**Genotypes**	n = 80	%	n = 81 *	%		
CC	56	0.70	51	0.63	ns ^§^	-
CT	5	0.06	21	0.26	0.0006	0.19
TT	19	0.24	9	0.11	0.0344	2.49
**Alleles**	2n = 160	%	2n = 162	%		
C	117	0.73	123	0.76	ns ^§^	-
T	43	0.27	39	0.24	ns ^§^	-

* rs1137070 available in one more healthy control. ^§^ ns = not significant.

**Table 4 diseases-12-00070-t004:** Allele and Genotype Frequencies of MAOA Gene (SNPs rs6323 and rs1137070) of Aggressive ADHD Patients in comparison to Non-Aggressive ADHD Patients and Controls.

	Aggress. ADHD		Non-Aggress. ADHD		Controls	
**MAOA SNP rs6323 (G891T)**						
**Genotypes**	n = 32	%	n = 48	%	n = 80	%
TT	22	0.70	34	0.71	51	0.64
GT	3	0.10	1	0.02	21	0.26
GG	7	0.20	13	0.27	8	0.10
**Alleles**	2n = 64	%	2n = 96	%	2n = 160	%
T	47	0.74	69	0.77	123	0.77
G	17	0.26	27	0.23	37	0.23
**MAOA SNP rs1137070 (T1410C)**						
**Genotypes**	n = 32	%	n = 48	%	n = 81 *	%
CC	22	0.70	34	0.71	51	0.63
CT	3	0.10	3	0.06	21	0.26
TT	7	0.20	11	0.23	9	0.11
**Alleles**	2n = 64	%	2n = 96	%	2n = 162	%
C	47	0.73	71	0.74	123	0.76
T	17	0.27	25	0.26	39	0.24

* rs1137070 available in one more healthy control.

**Table 5 diseases-12-00070-t005:** Allele and genotype frequencies of the Serotonin Transporter (5-HTT) gene polymorphism in Aggressive ADHD and Non-Aggressive ADHD patients.

	Aggressive ADHD		Non-Aggressive ADHD	
**5HTT SNP**				
**rs4795541(S)/(L) Polymorphism**				
**Genotypes**	n = 32	%	n = 48	%
LL	11	0.70	11	0.23
LS	16	0.10	28	0.58
SS	5	0.20	9	0.18
**Alleles**	2n = 64	%	2n = 96	%
L	38	0.60	50	0.52
S	26	0.40	46	0.48

## Data Availability

All data, code, and materials used in the analysis are available to any researcher for purposes of reproducing or extending the analysis and are available in the main text.

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
