# Peer review of "Aggressiveness in Italian Children with ADHD: MAOA Gene Polymorphism Involvement"

_diseases, 2024, doi:10.3390/diseases12040070_

Round 1

Reviewer 1 Report

Comments and Suggestions for Authors

This paper demonstrated that the aggressiveness in Italian children with ADHD may be involved with certain MAOA gene polymorphisms. The presentation is acceptable and the findings are useful. I recommended acceptance after some revisions.

1. The introduction section was too simple. Please enrich the content of this part. The importance of your study should be focused.

2. The captions of Tables were missing. Please add.

3. The format of Tables should be unified.

4. More detailed data should be presented in conclusion section.

5. The format of references should be checked and unified based on the requirement of this journal.

Author Response

Author reply: We thank the Reviewer for having appreciated our article and for the positive comments. According to her/his suggestions, we provided to revise the manuscript point by point, as suggested.

Reviewer

  1. The introduction section was too simple. Please enrich the content of this part. The importance of your study should be focused.

Author reply:

The introduction section was enriched and the importance of the study was better focused (All changes are written in red ink and marked in yellow).

Reviewer

  1. The captions of Tables were missing. Please add.

Author reply:

We provided to add the caption of all tables.

Reviewer

  1. The format of Tables should be unified.

Author reply:

The format of Tables was unified.

Reviewer

  1. More detailed data should be presented in conclusion section.

Author reply:

We provided to enrich the conclusion section (Lines 411-424).

Reviewer

  1. The format of references should be checked and unified based on the requirement of this journal.

Author reply:

References have been checked and unified following “Disease” requirements.

Reviewer 2 Report

Comments and Suggestions for Authors

The authors investigated the role of MOA gene polymorphisms on aggressiveness of children with ADHD.

-Did the authors calculate the sample size a priori? please provide details.

-Why there were more male participants included?

-Please provide demographic characteristics of subjects. 

-Please provide info on the scores used to diagnose ADHD and aggressiveness. Please clarify what were the characteristics adopted for aggressive behavior.  

-The statistical analysis was based on measures of association, however an analysis of regression of predictive modeling might be more suitable to determine  the influence of each polymorphism.

 -Please discuss about any prospective link with your findings and neurophysiological studies involving electrophysiological data (10.3390/ijerph19105953)

Author Response

Author reply: We thank the Reviewer for the comments; changes details are as following.

Reviewer

-Did the authors calculate the sample size a priori? please provide details

Author reply:

We thank the Reviewer for this observation. We provided details in lines 184-190.

Reviewer

-Why there were more male participants included?

Author reply:

We thank the Reviewer for pointing this out. In the period of enrollement, the majority of children with ADHD referred to the clinic were males.

Reviewer

-Please provide demographic characteristics of subjects. 

Author reply:

We provided to add these informations in Table 1.

Reviewer

-Please provide info on the scores used to diagnose ADHD and aggressiveness. Please clarify what were the characteristics adopted for aggressive behavior.  

Author reply:

We thank the Revisor for this comment; we provided all requested informations that previously are lacking (Lines 110-127).

Reviewer

-The statistical analysis was based on measures of association, however an analysis of regression of predictive modeling might be more suitable to determine  the influence of each polymorphism.

Author reply:We enriched the statistical analysis section with the lacking informations in lines 184-190. Results have been included in lines 261-263.

Reviewer

-Please discuss about any prospective link with your findings and neurophysiological studies involving electrophysiological data (10.3390/ijerph19105953)

Author reply:

We thank the Reviewer to offer us the possibility to discuss on a potential link among our study and neurophysiological studies involving electrophysiological data. In this respect, we included our considerations in  the Conclusion section (Lines 411-424).

Reviewer 3 Report

Comments and Suggestions for Authors

This manuscript analyzes the correlation between gene variation and the risk of aggressiveness in ADHD Italian children. Some suggestions are as follows:

Introduction:

Lines 44–76. This final paragraph appears overly extended, encompassing diverse but pertinent information. To enhance clarity and coherence, it is recommended to partition this paragraph into distinct segments, each addressing a specific aspect of the study's background and significance.

Discussion:

  1. The manuscript presents intriguing findings concerning the correlation between gene variation and aggressiveness risk in Italian children with ADHD. While the study is commendable for its scope, the presentation of results from multiple subjects warrants careful organization. Regarding Figure 1, its inclusion in the discussion does not notably enhance the manuscript's quality. Thus, its omission may streamline the narrative without sacrificing content.
  2. The discussion would benefit from a balanced interpretation of both significant and nonsignificant results. Authors are encouraged to elucidate the implications of each, delineating their respective contributions to the study's overarching objectives. Rather than resembling an introductory passage, the initial segment of the discussion should transition seamlessly into an in-depth analysis of the study's findings. For instance, while the manuscript briefly touches upon the involvement of the cingulate cortex, a more comprehensive exploration of the brain nuclei influenced by the studied genes, and their role in modulating aggressive behavior, is warranted. Such a discussion would enrich the manuscript and strengthen its scientific contributions.
Comments on the Quality of English Language

Brief typographical errors only.

Author Response

Author reply:

We thank the Reviewer for his/her comments that gave us the possibility to improve some parts of our manuscript. Manuscript revisions are explained point by point.

Reviewer

Lines 44–76. This final paragraph appears overly extended, encompassing diverse but pertinent information. To enhance clarity and coherence, it is recommended to partition this paragraph into distinct segments, each addressing a specific aspect of the study's background and significance

Author reply

According with this suggestion,  introduction has been structured  in 3 subsections to enhance clarity of the study. All changes are written in red ink and marked in yellow.

Reviewer

Discussion:

1.The manuscript presents intriguing findings concerning the correlation between gene variation and aggressiveness risk in Italian children with ADHD. While the study is commendable for its scope, the presentation of results from multiple subjects warrants careful organization. Regarding Figure 1, its inclusion in the discussion does not notably enhance the manuscript's quality. Thus, its omission may streamline the narrative without sacrificing content.

2.The discussion would benefit from a balanced interpretation of both significant and non significant results. Authors are encouraged to elucidate the implications of each, delineating their respective contributions to the study's overarching objectives. Rather than resembling an introductory passage, the initial segment of the discussion should transition seamlessly into an in-depth analysis of the study's findings. For instance, while the manuscript briefly touches upon the involvement of the cingulate cortex, a more comprehensive exploration of the brain nuclei influenced by the studied genes, and their role in modulating aggressive behavior, is warranted. Such a discussion would enrich the manuscript and strengthen its scientific contributions

Author reply

We welcomed the Reviewer's suggestions and provided to carefully revise the discussion to improve and clarify it. Also, we provided a context for the results of the study (Reply 1) (Lines 269-278; 280-281;316-333;349-353). Anyway, to be able to satisfy also the comments of the other reviewers, we decided not to upset the previous layout of the discussion and to keep the Figure 1.(Reply 2).

Round 2

Reviewer 2 Report

Comments and Suggestions for Authors

The paper is more suitable now. I recommend the authors to provide a table with mean values of Conners’ Rating Scales for male and female partocipants.

Author Response

We thank the Reviewer for the comments and according to her/his suggestions, we provided to introduce changes in the text (Lines 127-128 ) and a Supplementary Table  (Table S1) with mean values of Conners’ Rating Scales for boys and girls.